# Chronic Ionizing Radiation of Plants: An Evolutionary Factor from Direct Damage to Non-Target Effects

**DOI:** 10.3390/plants12051178

**Published:** 2023-03-04

**Authors:** Gustavo Turqueto Duarte, Polina Yu. Volkova, Fabricio Fiengo Perez, Nele Horemans

**Affiliations:** 1Belgian Nuclear Research Centre—SCK CEN, 2400 Mol, Belgium; 2Independent Researcher, 2440 Geel, Belgium; 3Centre for Environmental Sciences, Hasselt University, Agoralaan Building D, 3590 Diepenbeek, Belgium

**Keywords:** chronic exposure, acute exposure, nuclear accident, DNA damage, repair pathway, antioxidants, hormesis, trade-off, plant terrestrialization, plant evolution

## Abstract

In present times, the levels of ionizing radiation (IR) on the surface of Earth are relatively low, posing no high challenges for the survival of contemporary life forms. IR derives from natural sources and naturally occurring radioactive materials (NORM), the nuclear industry, medical applications, and as a result of radiation disasters or nuclear tests. In the current review, we discuss modern sources of radioactivity, its direct and indirect effects on different plant species, and the scope of the radiation protection of plants. We present an overview of the molecular mechanisms of radiation responses in plants, which leads to a tempting conjecture of the evolutionary role of IR as a limiting factor for land colonization and plant diversification rates. The hypothesis-driven analysis of available plant genomic data suggests an overall DNA repair gene families’ depletion in land plants compared to ancestral groups, which overlaps with a decrease in levels of radiation exposure on the surface of Earth millions of years ago. The potential contribution of chronic IR as an evolutionary factor in combination with other environmental factors is discussed.

## 1. Introduction

Most of the attention given to ionizing radiation (IR) is focused on its potential hazard for living organisms. However, IR’s importance goes further beyond this, comprising the formation of organic molecules [1], shaping habitable planetary environments [2], acting as a stress factor for entire ecosystems [3], and as an evolutionary factor. In this review, we outline the sources of IR, both as a natural component of the environment (Section 2) and due to anthropogenic activities following the growing demands of nuclear technologies (Section 3). The human-caused increase in IR’s environmental levels has direct consequences for ecosystems, which require effective countermeasures for the radiation protection of biota [4]. However, plants fall into a wide range of possible lethal doses, starting from 3–4 Gy of γ-radiation for seedlings of some crops and pine trees and exceeding 1000 Gy for clover seeds [5]. Nevertheless, for radiation protection, only three species are considered representative of all ecosystems: reference pine tree, reference grass, and reference algae [4]. In this context, we discuss the limitations and perspectives of the current radiation protection recommendations for plants (Section 3), considering the high variability of plant radiosensitivity across taxa [6]. Indeed, several factors contribute to species-specific radiosensitivity, such as the amount of genetic material and nuclear volume [7], the activity of transposable elements, and the DNA repair system [8], among others. The survival of an IR-exposed organism is determined by the damage severity (Section 4) and by the efficiency of the defence machinery to counteract the deleterious effects (Section 5). Interestingly, lower non-lethal doses of radiation exposure do not necessarily lead to an increase in mortality, instead triggering a range of possibly transient non-targeted effects, including growth and development stimulation after irradiation [9]. The mechanisms behind non-targeted effects are only starting to be unveiled (Section 6), and depend on the plant species, ionizing radiation type, dose and dose rate, ontogeny stage, season, and confounding environmental factors. 

Finally, approaching the effects and responses to IR from a different perspective, we highlight that DNA repair elements and pathways are evolutionarily conserved among *Archaea*, *Bacteria*, and *Eukarya* (Section 7). This observation suggests that the maintenance of genomic information and cellular integrity are ancient limiting factors for life viability. The frequency of mutations caused by IR exposure is dose-dependent [10]. Because of that, the significantly higher IR background levels on ancient Earth [11] may have demanded a higher input from the cellular defence systems for homeostasis maintenance, while surface IR levels would impose a physical limitation for life expansion. Starting from this hypothesis, we discuss how a possible trade-off between overall metabolic processes and the energetically costly damage protection mechanisms may have limited evolutionary pace and the colonization of land. In order to support this hypothesis, we compared the evolution of DNA repair gene families in *Viridiplantae* (“green plants”) (Section 7).

## 2. Natural Sources of Ionizing Radiation

IR is characterized by high-energy particles or waves capable of ionizing atoms and molecules. It derives from natural sources, such as cosmic rays (particulate and electromagnetic) or radioactive minerals in soil [11]. Cosmic rays are highly accelerated subatomic particles, which can be classified as solar or galactic. Although high-energy ultraviolet (UV-C) rays may cause the ionization of molecules, they are filtered by the atmosphere of our planet, not reaching the surface. Approximately 90% of the cosmic rays that reach Earth are galactic in origin, being composed mainly of protons (up to 87%), but also alpha particles (helium nuclei), and a small fraction of heavy ions and electrons [11,12,13]. Solar cosmic rays are ejected from the Sun and have lower energies, being composed of up to 98% protons and 2% by alpha particles [11,12]. Upon reaching Earth, the cosmic rays interact with the gaseous and particulate elements of the atmosphere, producing radioisotopes (e.g., ^7^Be, ^36^Cl, ^3^H, and ^14^C) because of the spallation reactions [12]. As a consequence, the higher the distance to Earth’s surface, the higher the incidence of cosmic rays, which is estimated to be 30 times greater between 6–9 km above the ground [12]. The cosmic radiation is also partly screened by the magnetic field of the planet [14], being one of the reasons why IR levels on our planet’s surface changed during Earth’s evolution, thus directly impacting the evolution of life [15].

The ionization of the atmosphere (and interstellar clouds) by cosmic rays are expected to generate molecules thought to be relevant to the origin of life [1,2,11,16]. For instance, boron is a cosmic ray-derived element that is essential for biochemical reactions. In cyanobacteria, boron is required for nitrogen fixation and has been implied in the evolution of photosynthetic organisms [17,18]. In plants, boron is a micronutrient that is important for cell wall formation, membranes, cytoskeleton, and cell division, thus supporting plant growth and development. Although it may cause DNA damage if in excess [19,20,21,22,23,24], its potential ability as a molecule stabilizer may have contributed to the emergence of life in the prebiotic context of the RNA world [18,25,26]. 

The crust of our planet contains natural radionuclides, which are classified as either primordial or secondary. The primordial radionuclides have been formed before the solar system, but due to their extremely long half-lives (>10^8^ years), they are still present on Earth, having decayed little since their formation [12]. They comprise the more common ^40^K, which is a β- and γ-emitter with a half-life of 1.26 × 10^8^ years, to rarer elements such as ^123^Te, an X-ray emitter with a half-life of 1.2 × 10^13^ years, and ^209^Bi, an α-emitter with a half-life of 2 × 10^19^ years [12]. The secondary radionuclides have half-lives from a fraction of a second to millions of years and are formed by the radioactive decay of the primordial ones. For instance, the decay of uranium (^238^U and ^235^U, half-lives of 4.5 × 10^9^ and 7.0 × 10^8^ years, respectively) to stable lead (^206^Pb or ^207^Pb), originates radioactive elements such as thorium (^227^Th, ^230^Th, ^231^Th, or ^234^Th), radium (^223^Ra or ^226^Ra), radon (^219^Rn or ^222^Rn), and polonium (^210^Po, ^211^Po, ^214^Po, ^215^Po, or ^218^Po) through the decay pathway, while releasing α and β particles [12]. 

At present, the average human population IR exposure is approximately 2.4 mSv/year (0.27 μSv/h) [12]. Yet, certain areas on Earth have higher background levels of IR as, for instance, Ramsar in Iran (30 μSv/h), Morro do Ferro and Guarapari in Brazil (22 and 20 μSv/h, respectively), and Mombasa in Kenya (12 μSv/h) [27,28]. For some plant species, chronic exposure due to natural radionuclides in soil has been investigated as a genotoxic factor [29,30]. In the oceans, naturally high radioactivity levels were also identified in marine communities living near hydrothermal vent ecosystems, where radionuclides derive from magma sources due to the tectonic activity [31,32]. For instance, in the East Pacific Rise and the Mid-Atlantic Ridge vents, high levels of uranium (^238^U, ^235^U, and ^234^U) and its daughter isotopes have been detected in various metazoans [32]. Although these abyssal food web communities are established below the euphotic zone and do not depend on plants or algae, hydrothermal activity has a direct impact on phytoplankton in the ocean [33]. Nevertheless, it remains to be investigated whether a similar effect due to the release of hydrothermal radionuclides in these areas would also affect the phytoplankton communities and the food chain as a consequence.

## 3. Anthropogenic IR Impact on the Environment and Plant Biota

Radioactively contaminated areas are increasing because of the anthropogenic impact on the environment, raising concerns about the radioprotection of humans and biota. Radioactivity is used as a source for nuclear medicine and power generation, being considered an alternative to the necessary low-carbon electricity sources [34,35]. Furthermore, IR has also been used as a mutagen for plant breeding, as a food sterilization agent, and also for delaying the germination, sprouting, and ripening of fruits and vegetables [36,37]. However, these areas of industry produce nuclear waste, which requires special management and infrastructure for long-term storage [38]. Although the radioactive decay of radionuclides present on Earth accounts for 50% of the dose received by the human population [12,39], uncontrolled radioactive emissions pose an immediate risk because of acute exposure of organisms to high doses, but also due to the long-term contamination of ecosystems. In this regard, most of the environmental contamination today derives from the nuclear fallout of bomb tests that occurred during the nuclear arms race after 1945. By 1996, when the Comprehensive Nuclear-Test-Ban Treaty was signed, 2052 nuclear tests had been performed—25% of which were in the atmosphere—mostly by the United States and followed by the former Soviet Union [40]. These tests released large amounts of ^14^C, ^137^Cs, ^90^Sr, ^131^I, and ^241^Am into the atmosphere, which were also incorporated by marine environments [41]. For instance, ocean sediments from Bikini Atoll in the Marshall Islands have been severely impacted due to the aboveground US H-bomb test (Castle Bravo) in 1954, and show high concentrations of ^239,240^Pu, ^241^Am, and ^207^Bi more than 60 years after the event [42].

Yet the most well-known events in terms of radioactive contamination were nuclear accidents. Recent reviews detail the effects of irradiation on non-human biota after Kyshtym (Russia, 1957) [43], Chernobyl (Ukraine, 1986) [44], and Fukushima accidents (Japan, 2011) [45]. Overall, the biological effects of nuclear accidents on non-human biota are different in the early phase (high-dose acute irradiation) and later phase, when short-living radionuclides decay and long-living radionuclides become responsible for the chronic IR exposure, representing the main input to the absorbed dose. For the radiobiological responses of plants, the period of irradiation is of utmost importance, since the plants’ IR sensitivity depends on the developmental stage [46]. During the seasonal growth period, plants are most sensitive and vulnerable to radiation-triggered DNA damage due to high meristematic activity and water availability, while physiological dormancy reduces the radiosensitivity during winter. Therefore the effects of radioactive fallouts on plant populations largely depend on the season of irradiation [3]. The Kyshtym accident happened in late September, and the winter dormancy period that followed mitigated the severity of the acute radiation damage [43]. Conversely, the Chernobyl disaster in late April led to dramatic damage to sensitive gymnosperm ecosystems (“Red Forest”), and morphological abnormalities were visible for different plant species [3]. The Fukushima disaster also happened in early spring, but no significant large-scale radiation damage occurred in plants in the exposed zone [45], although morphological anomalies were found in gymnosperm species [47,48,49]. Such a difference between Chernobyl and Fukushima disasters may be attributed to the composition and amount of ejected radioactivity, which at Chernobyl was an order of magnitude higher than at Fukushima [50].

The contaminated sites are, nevertheless, valuable sources for radiobiological research, especially for studying the effects of chronic radiation exposure on biota *in natura*. Unlike humans, the biota cannot be reallocated in the event of a nuclear accident or fallout, which renders them chronically exposed to the radioactive waste in their habitat. Historically, radiation protection has been human-centric, and because of that it is well-established. Conversely, the protection of the environment and wild species has only started developing in recent years [4,34,51,52]. Several difficulties permeate the comparison of biota response patterns in contaminated areas because each ecosystem is unique, and the contamination composition, weather conditions, and terrains are heterogeneous among the different sites. Some studies in such areas lack detailed information about the environmental parameters and dosimetry, thus hampering a full assessment of their conclusions. Furthermore, radiosensitivity is species-specific, depending on the chromosome volume and amount of photosynthetic material, among other structural and functional traits of each species [6,53,54]. For instance, while herbaceous plants are in general twice less sensitive to IR than woody trees, conifers are as hypersensitive as mammals [4,6,7]. However, under the scope of radiation protection, all plants are overall regarded as a less sensitive group, with three references: “pine tree—*Pinacea* family” (for large terrestrial plants), “wild grass—*Poaceae* family” (for small terrestrial plants), and “brown seaweed” (for aquatic plants). As a result, there is still no solid scientific consensus for the protection of non-human biota, and further focused investigation is necessary [34].

## 4. Cellular Effects of IR Exposure

The consequences of IR exposure to living cells depend on several factors: the source, type of radiation, dose rate, absorbed dose, and ultimately the radiosensitivity of the species (Figure 1). Although nuclear DNA damage is regarded as the primary direct consequence of IR exposure, mitochondria and chloroplasts as organelles containing electron transport chains and their own DNA also suffer similar detrimental effects of IR [55,56,57,58]. Because of the deposition of radiation energy onto nucleic acid molecules, unstable DNA radical cations are produced and undergo degradation, leading to the break of the phosphodiester bonds, and causing base damage [59]. The direct disruption of the DNA chain by single- and especially double-strand breaks (SSBs and DSBs, respectively) are the most serious damage caused to cells by IR exposure, leading to problems in chromatin organization, transcription, and replication, thus impacting essential molecular processes and cell functionality [60]. Direct IR-induced base damage comprises oxidation (e.g., guanine → 8-oxo-7,8-dihydroguanine, which mispairs with adenine leading to G-to-T mutations), substitution, or base loss [60,61,62].

The high water composition of most plant cells suggests that a significant part of the effects triggered by IR exposure may be indirect (Figure 1 and Figure 2) [63]. Water radiolysis occurs upon ionization of H_2_O molecules, which produces hydrated free electrons (e^−^_aq_), free radicals (H^•^, HO^•^, and HO_2_^•^), and their recombination products (OH^−^, H_3_O^+^, H_2_, and H_2_O_2_; Figure 2) [64]. However the generation of these reactive oxygen species (ROS) is also innate to cell metabolism, deriving from electron transport chains and as by-products of cellular processes in the chloroplasts, mitochondria, peroxisomes, apoplasts, plasma membranes, endoplasmic reticulum, and cell wall [65,66]. Free radicals and ROS accumulation cause oxidative stress, which leads to DNA base modifications (e.g., oxidation or deletion) and SSBs, amino acid modification and protein primary structure fragmentation, and lipid peroxidation [62,65,67]. Among them, hydroxyl radicals are extremely reactive and can damage nearly all molecules within a cell [65]. Furthermore, the formation of DNA apyrimidinic/apurinic (AP) sites can lead to the covalent linkage of the DNA with nearby proteins [68]. As a result, there is a disruption of the homeostasis of membranes fluidity, ion transport, enzyme activity, protein synthesis and cross-linking, and DNA integrity resulting, finally, in cell death [65,67]. Nevertheless, at the same time, ROS are important signalling molecules involved in hormone crosstalk, controlling stress responses and plant development from germination to senescence [66,69,70]. This means that the modulation of ROS levels depends on a fine-tuning between ROS generation and detoxification, which is severely unbalanced upon IR exposure. Finally, as DNA is complexed with proteins within the cells, IR also causes direct or indirect chromatin damage (Figure 1). The unstable protein and DNA radicals can recombine and form covalent bonds, leading to DNA intra- or inter-strand crosslinks, or DNA-protein crosslinks (e.g., actin, histones, and transcription factors) [59,68,71]. Crosslinks prevent DNA strand separation, thus hampering cell homeostasis and genomic integrity by blocking transcription and replication [68,71]. Furthermore, the inter-strand crosslinks repair process itself may produce DSBs [71].

## 5. IR Defence System

Several mechanisms and pathways are responsible for the cellular response to direct and indirect IR damage, and they have been extensively reviewed [36,60,62,72,73,74,75]. Furthermore, non-IR damage such as that caused by UV-A and UV-B can be repaired by the photoreactivation pathway, a conserved system found from *Archaea* to *Metazoa* except in placental mammals [76,77]. During photoreactivation, photolyases use light energy for reversing in situ base damage [36], but as a non-IR response, this pathway will not be detailed in this review. Most IR damage response pathways are also involved in the repair of errors introduced during the replication process (e.g., base mismatch, insertion, deletion, and strand break). Therefore, the proteins involved in these pathways are conserved across life domains, reflecting their ancient origin and fundamental importance. IR-induced cell damage and DNA lesions are sensed by specific machinery that can delay the cell-cycle progression for repairing the damage, induce apoptosis in the case of inefficient repair response or, in plants, promote endoreduplication to avoid DNA-damaged cell proliferation [78]. In parallel, cells have to modulate metabolic ROS generation and the activity of the antioxidant system to contain the IR-triggered oxidative stress. Although extranuclear effects of IR have been studied to a lesser extent, shifts in the redox balance have been detected in irradiated plants days after exposure, suggesting the involvement of the ROS-producing machinery on the homeostasis regulation [79]. Changes on photochemical quantum yield have also been reported after the exposure of photosynthetic organisms to IR [80,81,82,83,84,85]. Interestingly it has been found in mammalian cells that the mitochondrial DNA is significantly more susceptible than its nuclear counterpart to both IR-induced oxidative damage and DNA strand breaks [55,86,87,88]. This may be due to the absence of histone protection and an efficient DNA repair mechanism in this organelle [55,89]. Nevertheless, as ROS sources, both mitochondria and chloroplasts have damage defence systems, sharing some factors with the nucleus [90]. Organellar genomes do not encode DNA repair enzymes and have to import nuclear-encoded ones from the cytosol [90]. For instance, it is known that the modulation of antioxidant levels, photoreactivation, homologous recombination (HR), strand break repair, and a modified base excision mechanism are involved in chloroplastic and/or mitochondrial DNA protection, as reviewed elsewhere [90,91]. Interestingly it has been suggested that lower radiosensitivity levels may be partially linked to the presence of chloroplasts and chlorophyll content [92,93]. At the end, the outcome of IR exposure is a consequence of the interplay between the damage severity and the defence machinery efficiency.

### 5.1. DNA Repair Mechanisms

IR-induced DNA base damage can be repaired via nucleotide excision repair (NER), base excision repair (BER), and mismatch repair (MMR) pathways [36,72,74,94]. BER’s primary function is to remove small non-helix-distorting base lesions, such as AP sites, oxidized or hydrated bases, but also SSBs [72,74,95]. In summary, a base-specific DNA glycosylase (e.g., OGG1) recognizes and removes the damaged site creating an AP site, which is cleaved by an AP-endonuclease at the phosphodiester bond. Depending on the nature of the lesion, the repair is performed either by a “short” or “long” patching mechanism. In mammals, the short-patching BER relies on DNA polymerase β, XRCC1 (X-ray repair cross-complementing protein 1) and/or PARP1 (Poly(ADP)ribose polymerase 1), and DNA ligase 3α for fixing a one-nucleotide gap [95,96]. The long-patching pathway removes ~10 nucleotides around the lesion and depends on DNA polymerase β or δ and DNA ligase I for completing the repair process [95]. While homologues to several mammalian BER-related genes have been identified in plants, the apparent lack of Pol β and DNA ligase 3 implies that their functions may be performed by other proteins [72,74,75,97].

NER is a very conserved pathway that recognizes steric (conformational) changes in DNA helix structure or base dimers for the bulky excision of a 24–32 base oligonucleotide containing the damaged site, following the DNA synthesis in the single-stranded region [74,75]. NER is induced by two different damage recognition systems that use the same repair machinery. The DNA lesion recognition by RNA polymerase II causes its stalling during transcription, which triggers the recruitment of Cockayne Syndrome B protein (CSB). The interaction with Damage-specific DNA Binding protein1 (DDB1)–CULLIN4 (CUL4) E4 ubiquitin ligase–CSA protein complex initiates the transcription-coupled repair (TC-NER/TCR) process [75]. Damaged but non-actively expressed genomic regions are detected by the DDB2–DDB1–CUL4 complex for the global genome repair (GG-NER/GGR), which recruits *xeroderma pigmentosun* group C (XPC), Radiation Sensitive 23 (RAD23), and CENTRIN protein complex [74,75]. Next TC- and GG-NER converge. The transcription elongation factor-IIH (TFIIH) complex unwinds the lesion site, while the helicases XPB and XPD nick both sides of the damaged strand region for its removal, following strand synthesis by an X family DNA polymerase [74,75]. In mammals, this task is performed by DNA Pol δ, whereas in *Arabidopsis thaliana* it may be performed by AtPol λ [74,75].

The MMR pathway’s primary function is to correct errors introduced during DNA replication as it recognizes and fixes mismatches in DNA strands. Hence, it is also fundamental for correcting mispairing due to IR exposure, playing an important role in plants’ meristematic cells. As for NER, MMR is a conserved pathway among bacteria, yeast, metazoans, and plants, although the latter show duplicated copies of MMR genes in their genomes [75]. In plants, MSH and MLH are heterodimeric ATPases with functional homology to the homodimeric bacteria MutS and MutL, respectively. MutS/MSH recognizes base–base mismatch or insertions/deletions, and the interaction with MutL/MLH leads to the assembly of the DNA repair complex and recruitment of MutH, an endonuclease that nicks the unmethylated DNA strand. Next, PCNA, DNA helicases II, exonuclease 1 (EXO1), RPA, and DNA polymerases (Pol δ) perform the correction of the damaged site [74,75].

Homologous recombination (HR) and non-homologous end-joining (NHEJ) repair pathways are capable of circumventing DSBs, which are the most harmful IR-induced damage to cells [36,74]. During the NHEJ, the broken DNA strands are stabilized and re-ligated, which often results in mutations or base losses. HR is a conservative mechanism that uses homologous sequences as templates for repair. In plants, animals, and yeast, the primary signal transducers for damage repair upon DNA breakage are Ataxia Telangiectasia Mutated (ATM) and ATM and Rad3-related (ATR) protein kinases. ATM and ATR play both distinct and additive roles, whereas the former mainly responds to DSBs, and the latter is activated by SSBs. In summary, ATM activation relies on the MRE11, RAD50, and NBS1 (MRN) complex that recognizes DSBs [72,98]. The MRN complex degrades the 3′ end, and leads to the accumulation of single-stranded DNA coated with Replication Protein A (RPA), which protects the damaged chain from degradation, and is recognized by ATR Interacting Protein (ATRIP), which recruits ATR for its activation by the 9–1–1 complex (i.e., RAD9, RAD1, and HUS1) [99]. Both ATM and ATR signalling networks converge to Suppressor of Gamma-response 1 (SOG1), a transcription factor with a similar function as the metazoan p53 transcription factor, which controls the activity of several DNA repair and cell cycle genes [73,100]. SOG1 along with RAD54 are required for chromatin mobility during HR in plants, which is hypothesized to increase the encounter chance between the damaged site and the repair template [101]. Nevertheless evidence also supports the existence of a SOG1-independent pathway in plants via the Retinoblastoma-related (RBR1) hub and repair involving RAD51 and Breast Cancer Susceptibility 1 (BRCA1) [73,102,103]. Interestingly, it has been found that small RNAs are involved in DSB repair response via Argonaute 2 (AGO2), which is required for RAD51 recruitment [104,105]. Furthermore, the plant-specific CDKB1–CYCB1 (B1-type cyclin-dependent kinase and B1-type cyclin, respectively) complex is also necessary for the recruitment of RAD51 to the damaged DNA site for HR [106].

During the NHEJ, both strands can be rejoined with no or little processing. Because of that, it can lead to sequence mutations [60,75]. The core proteins of the canonical NHEJ pathway are the KU70/KU80 complex, which recruits DNA-dependent protein kinases for activating nucleases, DNA polymerases, and the XRCC4–Lig4 complex [75]. Alternatively, PARP proteins are involved in the repair by microhomology-mediated end joining (MMEJ), an error-prone repair process that does not require the involvement of KU proteins [74,75]. Although NHEJ is the most common DSB repair system in plants, the HR pathway may be preferred depending on the cell cycle and homologous repair template availability [74,75]. During the HR, 5′-3′ degradation (resection) at the damaged site occurs, exposing the single-stranded 3′-OH overhangs that can prime the DNA synthesis. If the overhangs are complementary they can anneal, resulting in the gap filling by DNA polymerase, deletion of overhangs, and ligation by DNA ligase I (single-strand annealing pathway; SSA), in a process mediated by RAD52 [60]. However this is a non-conservative mechanism that may lead to deletions within repeated (tandem) sequences. On the other hand, the invasion of the resected region by a homologous, double-stranded DNA template—in a process mediated by RAD51—leads to the synthesis of the complementary region (synthesis-dependent strand annealing; SDSA). This process may involve the formation of a Holliday junction, which results in the crossover between the pairing chromatids.

Crosslinks repair depends on factors such as their chemical structure, DNA sequence, their local dynamic structure, and flexibility [68,71]. Since the crosslink repair may require the formation of DSBs, the multifunctional MRN complex and the HR machinery also play a role in the process [68]. Although the majority of the crosslinks are the substrates for the NER pathway, IR-induced DNA–protein crosslink repair is limited by the size of the attached protein [59,68]. Furthermore, in the event of DNA–protein adduct formation, the repair strategy requires proteins with a specific proteolytic function for degrading the attached molecule, making the remaining peptide accessible for downstream repair mechanisms [68]. The general DNA–protein crosslink repair pathway in *A*. *thaliana* involves the metalloprotease weak suppressor of smt3 protein 1 (Wss1), which has first been identified in yeast [107]. In parallel, inter-strand DNA crosslinks may be repaired by inducing DSBs, triggering HR, and processing Holliday junctions through the MUS81–EME1 endonuclease complex [108,109]. Finally, a third pathway requiring tyrosyl-DNA phosphodiesterase 1 (TDP1) is responsible for fixing spontaneously occurring enzymatic DNA–protein crosslinks [68]. Interestingly, it has been found in mammals that PARP1, which also plays a role in the BER and NHEJ repairing processes, is involved in DNA–protein crosslink repair in an IR-specific way [110]. Furthermore, in *A*. *thaliana,* BRCC36A is epistatic to BRCA1 for HR and DNA crosslink repair [111].

### 5.2. Antioxidant Mechanisms

The antioxidant ROS scavenging system consists of enzymatic and non-enzymatic components. The enzymatic antioxidants are mainly represented by the ascorbate peroxidase (APX), catalase (CAT), glutathione peroxidase (GPX), superoxide dismutase (SOD), and thioredoxins (Trx) [66,67,69,112]. The main non-enzymatic components are ascorbate (ASC), glutathione (GSH), tocopherol, carotenoids, and proline [66,67,69,112]. The antioxidant system operates at different cell compartments, and specific antioxidants can be activated to respond in a stress-specific way [67]. Interestingly, the homeostasis of the antioxidant system is maintained by balancing the activity of the different enzymes, which relies on crosstalk among the different stress response and hormone pathways and which also requires a fine-tuning of their gene expression profiles [113,114,115]. The potential of antioxidants for improving stress resistance has been investigated in different plant species [67,113,115,116,117,118,119,120,121], and the status of the antioxidant system and the control of its genes’ expression is commonly used as a readout for assessing the impact of IR exposure in plants [79,85,122,123,124,125,126,127,128].

APX, GPX, and CAT are mainly H_2_O_2_ scavengers, but they rely on different mechanisms for catalysing the detoxification reaction. APXs use ASC as an electron donor for generating dehydroascorbate and water [115]. The dehydroascorbate is then recycled, maintaining a high ASC/dehydroascorbate ratio for an efficient redox pool [121]. Different APX isoforms have been specifically located in the chloroplast stroma, cytosol, microsome membrane, or thylakoids [115,129]. APXs also have a fundamental role in the ascorbate-glutathione cycle, which is the main hydrogen peroxide scavenging process in plants during stress responses [113]. CAT, on the other hand, is a heme-containing enzyme that mainly occurs in the peroxisomes but also in the cytosol and mitochondria and does not require a cellular reducing equivalent for catalysing the dismutation reaction [113,115]. It has high specificity to H_2_O_2_ but weak activity against organic peroxides [67]. CAT has a central role in photorespiration, but it is also involved in plant acclimation during stress responses [113]. GPX, on the other hand, uses GSH and thioredoxin as reducing agents for detoxifying H_2_O_2_ to H_2_O [113,114]. In plants, GPX has been identified in the chloroplast, mitochondria, cytoplasm, and nuclear and extracellular regions [114].

SOD is a metalloenzyme and efficient superoxide anion radical (O_2_^•−^) scavenger, catalysing its dismutation and thus reducing the formation risk of hydroxyl radicals (HO^•^) [115]. They are classified according to their metal cofactor, and show compartment specificity: copper/zinc (Cu/Zn-SODs) are found in the cytosol and chloroplast, iron (Fe-SODs) are present in the chloroplast, and manganese (Mn-SODs) locate in mitochondria and peroxisomes [120]. While Cu/Zn-SOD probably evolved separately in eukaryotes, Mn- and Fe-SOD are the more ancient types of SODs [130], whereas the evolution of different metal requirements is probably related to the soluble metal compounds and O_2_ content in the biosphere during different geological eras [131].

Trxs are ancient redox regulators found in prokaryotic and eukaryotic organisms. They catalyse the reversible disulphide-bond formation and are recycled by Trx reductases using NADPH or reduced ferredoxin as electron donors [132]. In plants, Trxs are involved in a complex network for signalling and cell homeostasis control [132]. For instance, approximately 40 Trxs and Trx-related genes have been identified in *A*. *thaliana*, encoding 20 different Trxs isoforms that are organelle-specific, and that are involved not only in oxidative stress response, but also in metabolite biosynthesis, electron transport chain, photosynthesis regulation, and in the Calvin–Benson cycle [132,133]. Furthermore the Trx system interplays with the GSH pathway being important for postembryonic development [133].

Among the non-enzymatic antioxidants, ASC and GSH are the major ROS scavengers in plants. ASC works as an electron donor for directly eliminating H_2_O_2_ via ascorbate peroxidases [67,134,135], while its accumulation in the apoplasts is thought to link the plant response to environmental stress [136]. Improving ASC biosynthesis has been reported to increase stress tolerance in different plant species [137,138,139,140], while its protective role against IR has long been known [141]. ASC is also involved in stress response as a cofactor for other enzymes, via the biosynthesis of hormones such as ethylene and abscisic acid (ABA) and epigenetic regulations [135]. GSH is synthesized in the cytosol, but it is present in basically all cell compartments [121]. With a strong reducing power, GSH can directly react with O_2_^•−^, HO^•^, and H_2_O_2_; it can protect proteins, lipids, and DNA by forming adducts with the reactive electrophiles; it may donate protons in the presence of ROS, resulting in the formation of glutathione disulphide [67]. The ascorbate–glutathione cycle links both antioxidants, where glutathione is involved in the regeneration of ascorbate from dehydroascorbate, and the ASC and GSH pools are balanced to control the homeostasis of the antioxidant system [67,121].

Tocopherols and carotenoids are lipophilic antioxidants. While the former are specific to photosynthetic organisms, the latter are also found in fungi and non-photosynthetic bacteria [117,142]. They can physically quench ROS, especially oxygen singlets, protecting lipids and membranes [117,142]. Tocopherols’ regeneration to their reduced form depends on ASC and GSH [116]. Carotenoids are also essential for the biosynthesis of ABA, which is a central hormone controlling plant development and stress responses [142]. Proline is an amino acid synthesized from glutamate and ornithine that may act as a ^1^O_2_, H_2_O_2,_ or HO^•^ quencher [118]. Although external proline supply under normal conditions may induce ROS accumulation and thus impair plant growth due to toxicity [143], under stress conditions it can increase the GSH pools, acting in parallel with the ASC pathway for controlling cell redox homeostasis [118,119].

## 6. From Acute High-Dose to Chronic Low-Dose IR Exposure: Non-Target Effects in Plants

High-dose IR exposure leads to toxic effects, because the direct and indirect damage exceeds the quenching limit of the organism’s defence systems. It can cause genetic and epigenetic changes, followed by the severe homeostatic disruption of cellular metabolism, which can manifest within seconds or decades after exposure [9,72]. On the other hand, low-dose irradiation may trigger a set of so-called non-target radiobiological effects.

Direct DNA damage is the main target effect of high-dose irradiation, where a direct interaction of a quantum or a particle with the genetic material leads to DNA strand breaks or other types of mutational events. Therefore, in theory, only a cell which contains at least one track of a gamma quantum or an ionizing particle should show direct DNA damage. However, non-irradiated cells of an irradiated organism exhibit responses similar to the directly irradiated ones, including DNA damage and ROS production [9]. Such phenomena are known as the non-target effects of ionizing radiation, and they are generally subdivided as radiation-induced bystander effect (RIBE), genomic instability, radiation hypersensitivity, radiation hormesis effect (RHE), radioadaptive, and transgenerational responses [144]. Most observations of the non-target radiation effects were made on humans and other mammals, while data on plants are scarce and rather descriptive, without a deep understanding of the underlying molecular mechanisms. However in recent years several mechanisms of RIBE and RHE occurrence after the irradiation of plants have been investigated.

RIBE occurs when low-dose radiation exposure of some cells affects non-irradiated neighbouring cells. Depending on the experimental setup, the bystander effect can be observed even at the organismal level, as was shown after X-ray and γ-irradiation of the radioresistant *A*. *thaliana* leading to an increase of HR frequency in neighbouring non-irradiated plants [145]. Similarly, the α-particle irradiation of the distal primary roots of *A*. *thaliana* seedlings resulted in a significant increase in the frequency of HR in the aerial parts [146]. The increased induction of HR occurred in every true leaf over the course of rosette development, followed by short-term upregulation of the HR-related *AtRAD54* gene in the non-irradiated aerial parts. Treatment with a ROS scavenger dramatically reduced those effects, suggesting that ROS play a critical role in mediating the bystander mutagenic effects in plants [146]. In rice, an IR response signature was identified in young non-irradiated leaves of γ-irradiated plants as a part of a systemic acquired acclimation, while the old irradiated leaves reached a new homeostasis status [128]. The involvement of ROS in RIBE was also shown in an experiment with young *A. thaliana* seedlings [147]. The local irradiation of *A. thaliana* roots with 10 Gy of α-particles led to long-distance changes in DNA methylation patterns, indicating the existence of RIBE-mediated epigenetic modifications in higher plants [148]. RIBE-promoted genome methylation machinery modulation also led to the transcriptional activation of mobile genetic elements and histone modifications [148]. Transcriptional activation was significantly inhibited by the treatment with an ROS scavenger, thus confirming a pivotal role of ROS in RIBE [149]. Interestingly, the simultaneous root-localized irradiation of *A. thaliana* plants and rotation in a clinostat showed that the simulated microgravity inhibited significantly the RIBE-mediated generation of ROS and upregulation of the expression of *AtRAD54* and *AtRAD51* genes, but made no difference to the induction of HR by RIBE. Such a divergent response was associated with the microgravity-triggered modulation of generation or translocation of the bystander signal(s) in roots [149]. The signalling was shown to be dependent on jasmonic acid (JA) [150]. Pre-treatment of *A. thaliana* seedlings with a JA biosynthesis inhibitor significantly suppressed the RIBE-mediated expression of *AtRAD54*, induction of HR-related genes, and epigenetic changes in the aerial parts of root-irradiated mutants deficient in JA biosynthesis (*aos*) and signalling cascades (*jar1-1*).

Another non-target effect of irradiation is radiation hormesis. RHE is a biphasic dose-response relationship reflecting plant growth, development, and stress tolerance improvement by low doses of IR and pronounced inhibition by high doses of IR. Hormetic responses after radiation exposure on plants were shown for many endpoints, plant species, and radiation types [151]. The hormetic effect is non-specific and can occur after low-intensity exposure to different stress factors. Several mechanisms possibly underlie RHE, including activation of heat shock proteins, proteasomes, kinase cascades, changes in nitrogen metabolism, phytohormonal balance, and, in general, repair and antioxidant response processes [152,153,154,155]. Enhanced growth and developmental rates after low-dose IR may be related to the overcompensation following the initial disruption in homeostasis triggered in response to the stress [156]. Specifically, DNA repair induction [157], changes in phytohormonal balance (especially related to abscisic acid and JA) [79,154], antioxidant capacity [158,159,160], and the stimulation of photosynthesis and pigment accumulation [161,162,163,164] can be among mechanisms responsible for RHE occurrence. Still, the production of ROS is considered a driving force behind RHE, making these molecules essential for the non-target effects of irradiation.

IR-induced genomic instability has been documented for various endpoints including chromosomal aberrations and mutations, which have also been reported in the descendants of irradiated mammalian or yeast cells many generations after the initial exposure [165]. In plants, this effect was seen as an increase in micronuclei frequency in the descendants of γ-irradiated tobacco cells, more than 22 generations after the irradiation event [165]. Long-living pine trees in the Chernobyl exclusion zone kept a high frequency of cytogenetic abnormalities decades after the initial acute IR exposure in 1986 [166], although at least partially this may also be attributed to the chronic radiation exposure.

Overall, the same signals that are involved in the direct response to the initial IR event (e.g., JA, abscisic acid, and ROS), may trigger different non-target effects. Furthermore, the degree of the effect seems to depend on the type of IR, dose and dose rate, plant species, ontogeny stage, and many other factors. Non-target effects are not stressor-specific, and as a consequence of IR, they are more evident after acute exposure. Nevertheless, they may also be studied on chronically irradiated plants in radioactively contaminated areas [85,166], although transcriptional responses to acute or chronic irradiation are rather different and often opposite [167,168,169].

## 7. DNA Damage Response to Chronic IR under an Evolutionary Perspective

DNA damage is ubiquitous. As described earlier, IR can directly or indirectly impair nucleic acids. Nonetheless, other abiotic stresses such as drought, salinity, heat, UV, and chemical agents are also potentially damaging sources [67,73,113,115]. Consequently life maintenance relies on efficient mechanisms that protect genetic information integrity. In the model plant *A. thaliana*, the mutation rate per site per generation has been estimated as 6.95 × 10^−9^ (SE ± 2.68 × 10^−10^) for nucleotides and 1.30 × 10^−9^ (±1.07 × 10^−10^) for indels [170]. These values reflect the high efficiency of the DNA protection machinery. 

Life is well-adapted to the considerably low IR levels on Earth’s surface at present, which means that in uncontaminated areas other stress sources are more relevant to living organisms, especially in the context of changing climates. However IR (and UV) levels were drastically different when life emerged on Earth. Not surprisingly, mechanisms controlling defence responses to a primordial stress factor such as DNA damage—and thus to IR exposure—must be highly conserved among *Archaea*, *Bacteria*, and *Eukarya*. Indeed the diversification of DNA protection pathways in the three life domains occurred from a minimal set of genes [171,172,173]. Yet, a general assumption is that primitive organisms are more radioresistant, while radiosensitivity varies across different species and phyla [53,174]. An interesting fact is that species can modulate their global genome repair processes in accordance with damaging stresses and cell replication status [175]. Such a requirement for modulation also indicates that the maintenance of an overactive DNA repair machinery must be energetically costly, with implications for the organism’s homeostasis. For instance, unless properly controlled, DNA repair activities may cause energy depletion in bacteria, leading to cell death [176], while the balance between oxidative stress resistance and DNA repair has a direct impact on mutation rates and evolution [177]. Therefore, under chronic IR exposure, there must be a trade-off between the activity of damage protection mechanisms and overall metabolic processes that impact growth and development. For photosynthesizing organisms, it implies a balance between the generation of energy (photosynthesis and respiration), ROS production (both as IR and metabolic by-products, and as signalling molecules), and damage scavenging elements activity. This observation has been previously noted based on the transcriptomic and enzymatic profiles of plants living under low-dose chronic IR exposure conditions, such as *Pinus sylvestris* and *Capsella bursa-pastoris* growing in contaminated areas of the Chernobyl exclusion zone [85,168]. Furthermore, it is also plausible to speculate that if an overactive and overprotective system was indeed required in past eons for maintaining the genetic integrity of the organisms in an inhospitable environment, the mutation rate per generation may have been lower, hence implying a slower life evolutionary rate during early Earth. In this context, an aspect that has been generally neglected is the potential contribution of chronic IR as an evolutionary factor that limited the emergence of more complex organisms, their diversification, and land colonization [15,178,179]. 

### 7.1. Early Earth Conditions: A Chronically Radioactive Environment

Evolutionary forces drive species adaptation, niche occupation, and the increase of life complexity. Yet, strict physical conditions limit life on Earth, namely temperature and pressure windows that allow water in its liquid state, nutrient-rich environments, shielding atmosphere yet adequate solar irradiation, and low IR levels [15,180]. While life origin is still debated [181,182,183,184,185], it is clear that the early Earth environment was challenging for the primordial organisms and that the evolution of systems capable of protecting the genetic information and transmitting it to the next generation was a critical step for the success of any species. Even though during Earth’s early years the Sun was much dimmer than it is now—its luminosity estimated to be up to 70% lower in a condition known as the Faint Young Sun paradox [186,187,188]—the incidence of IR on the planet’s surface was extremely high until the Cambrian. Solar high-energy particles could have been up to 1000 times stronger in the Archean than now, gradually decreasing during 2 billion years [189,190]. It has been estimated that because of the absence of a shielding atmosphere, UV exposure and solar cosmic rays incidence were 400 and 5 times higher at sea level, respectively, 4 billion years ago (Bya) [191]. Other cosmic episodes, such as supernovae and nebula encounters may have also contributed to high levels of galactic cosmic rays incidence on Earth, possibly accounting for species extinction events [192,193]. Coupled with weak periods of the Earth’s magnetic field, high-energy gamma-ray bursts may have also contributed to glaciation events known as the Snowball Earth periods between ca. 850 and 800 and 600 and 550 million years ago (Mya) [15,178]. In such an extreme environment, and before the emergence of more diverse and complex protection mechanisms, the primordial organisms probably depended on natural resources for avoiding deleterious damage, while the expansion to new environmental boundaries was limited. 

Water is a natural radiation shield, and not surprisingly for nearly 3.4 billion years (Archean to Cambrian) life on Earth was dominated by aquatic species. While multicellularity emerged at least 2 dozen times during the 2 billion years of *Eukarya* evolution [194,195], the oldest evidence of eukaryotes living in non-marine environments dates to the Proterozoic (Torridonian, 1.2-1.0  Bya), yet is associated to aquatic (freshwater) systems [196]. The oldest known *Chlorophyta* (a clade composed of algae that along with *Streptophyta* constitute the green plants [197]) is 1 billion years old [198]. An evolutionary conundrum is the question: why would life emerge in such a short time after Earth’s origin, but the evolution of more complex and diverse organisms that colonized the lands took so long to occur?

### 7.2. Plants Diversification and Terrestrialization: Was Chronic IR Exposure a Limiting Factor?

The colonization of the land by plants is one of the most important events of Earth’s evolution. It probably occurred as a step-wise process [199], starting with basal *Bryophyta* during the Early Palaeozoic, although it is still debated whether it was as a hornwort-, liverwort-, or moss-like organism [200]. This green revolution shaped the geosphere, impacting on weathering and soil fertility, outlining global climates and ecosystems [200,201,202,203], and possibly contributing to the burst of animal phyla that characterizes the Cambrian explosion [204,205]. Land plants (*Embryophyta*) probably evolved from the *Charophyta* algae linage in which seems to be a one-time event between the mid-Cambrian to lower Silurian (515–430 Mya) [206,207,208,209]. Later, embryophytes diversified into the major land plant lineages, including hornworts, liverworts, mosses, lycophytes, monilophytes, and spermatophytes [200,201,210]. It has been suggested that the distance between *Charophyta* and *Embryophyta* is small [211], and that the traits that allowed the plants’ terrestrialization were available before *Embryophyta* emerged [212], such as apical meristems, asymmetric cell division, branching patterns, and plasmodesmata [195,200]. It has also been proposed that symbiotic associations with fungi were fundamental for land colonization and the adaptive radiation of both groups [213]. Although fungal land colonization is a late event, per se, that occurred ca. 720 Mya, ancestral fungi must have also been aquatic, meaning that they were also protected from the ionizing terrestrial environment [214]. However, the fossil record dating to the origin of the embryophytes is scarce, only becoming abundant after they had diversified and covered the land [212,215]. Moreover, cryptospore assemblages suggest that the evolution rate of the embryophytes during their initial 35–45 million years was extremely slow [216]. These observations are especially interesting under the scope of the niche theory [217,218]. While this might reflect constraints limiting both land colonization and species radiation, they may also represent the punctuated equilibrium phenomenon.

Both terrestrialization and the subsequent species diversification became the hallmark of the Cambrian correlate, with a gradual change in Earth’s magnetosphere, which allowed the establishment of a filtering atmosphere (Figure 3). During early Archean, the magnetic field of the planet was between 25 and 50% of present-time levels and about 60% during late Archean (2.7 Bya), increasing by 40% during the last 570 million years [15,219,220,221,222]. The increase in Earth’s magnetic field must have contributed to deviating solar wind, allowing the atmosphere to grow and protecting the surface from high-energy particles [11,15,223,224,225]. The stronger magnetosphere would also allow the formation of the Earth’s ozone layer [15], which is a primary filter against high-energy particles and ionizing UV-C rays between 195 and 290 nm [188,226,227]. Interestingly, from an ecological perspective, the increase of the magnetosphere and atmosphere—thus the reduction of IR on the surface of the planet—represented the opening of niches to be colonized and explored. In parallel, such a reduction also allowed the ancient organisms to diminish the DNA protection mechanisms activity, which is energetically costly. That also means that these organisms would have more energetic resources for supporting an increase in complexity. The attenuation of the DNA protection system may be achieved by the constant repression of the overactive DNA repair and scavenging processes. Although that could have been a possible solution at early evolutionary stages after the environmental change, it still requires a significant energetic input. Considering an evolutionary long-term perspective, the alternative would be the reduction of the number of elements (genes) involved in the DNA repair and scavenging processes, which would lead to further energetic improvement for maintaining homeostasis. Indeed, although there might be dosage compensation in some cases, alterations in gene copy number are followed by proportional expression changes that contribute to phenotypic variation and consequently to adaptation and evolution [228,229,230]. Altogether, one would expect that among green plants (*Viridiplantae*), the most basal species must be enriched for genes related to damage protection, which would gradually decrease in more diverged organisms until reaching a homeostatic balanced level that is relevant for the contemporary stress levels.

For testing the plausibility of this hypothesis, we evaluated the expansion or depletion of gene families related to DNA repair processes across plant species, comparing the earliest to the most diverged groups. By doing so, sequence homology that would tend to group the most closely related species was not taken into account. This analysis is represented in Figure 4. When considering major *Viridiplantae* groups, it becomes evident that basal *Viridiplantae* (*Chlorophyta* + *Prasinodermophyta*) are enriched in genes related to DNA repair processes in comparison to land plants, while the basal land plants (*Lycopodiopsida* + *Anthocerotopsida* + *Marchantiopsida* + *Bryopsida*) represent an intermediate/transitory state (Figure 4A). Nevertheless we cannot exclude that cofounding factors and gain of function may also have contributed to the depletion of certain DNA repair gene families. It is interesting to note that certain gene families show a peak in basal land plants, such as HOM05D000001 (BRCA1-associated RING domain protein 1 family), HOM05D002009 (DNA polymerase delta subunit 1 family), and HOM05D004663 (uracil–DNA glycosylase) (Figure 4A; Appendix A). They may represent important functions for land colonization, before the emergence of more complex mechanisms for coping with land environmental stresses. The pairwise comparison between each *Streptophyta* and each basal *Viridiplantae* (*Chlamydomonas reinhardtii*, *Micromonas commode*, and *Prasinoderma coloniale*) highlights the overall DNA repair gene families’ depletion in land plants relative to the latter group (Figure 4B). Interestingly, the pairwise comparison also shows that certain gene families changed little from *Chlorophyta* to *Embryophyta* (white boxes), and additionally the presence of species-specific adaptations, which may reflect the involvement of repair processes on habitat stress responses that are unrelated to IR (e.g., drought/desiccation and heat stress).

## 8. Perspectives

In the context of the evolution of IR defence mechanisms in plants, it would be interesting to investigate the radiosensitivity and IR defence systems of plants such as *Lemnaceae*, which returned to aquatic environments after land colonization [236]. So far, DNA damage and repair pathways in these plants have not been examined in detail. Comparative genomics of such species may also provide insights into the driving mechanisms of land colonization. Furthermore, it may expand the current approaches for biodiversity preservation, since aquatic species of higher plants are not represented under the scope of radiation protection.

The functional and genomic adaptations to chronic radiation exposure are also relevant for plant performance in the context of increasing anthropogenic genotoxic activity. Because IR adaptive responses often require elements that are involved in different biotic and abiotic stress response pathways [85,168,169], γ-radiation could be used as a proxy for studying adaptation to other genotoxic agents (e.g., heavy metals, UV-C in space environment, various chemicals, etc.). Moreover, the similarity found between transcriptional profiles of plants exposed to chronic IR and those exposed to climatic stressors indicates that the strategies developed to cope with IR could potentially be utilized to enhance the resilience of crop species to various stresses, which is relevant in light of food security in future climates.

Finally, identifying the genomic determinants of radioresistance would also be valuable for future space exploration and colonization efforts. Because the re-supply of resources to these missions is a limiting factor, establishing cultures under a highly radioactive environment (i.e., outside Earth) will be necessary. Photoautotrophic organisms may be used as sources of nutrients and oxygen. However, it is essential to understand the long-term effects of ionizing radiation on plant growth, development, and metabolism, the factors that would directly impact the viability of a plant species cultivation in Space.

## 9. Conclusions

Under the human perspective, IR is often minimized to simple short-term cause–effect issues; however it has a complex interplay with non-human biota. Anthropogenic radionuclides’ release into the environment (i.e., nuclear energy, medicine, industry, and weapons) can permanently modify entire ecosystems. However understanding the impact and consequences of chronic irradiation on an ecological scale is far from ideal. Such a lack of knowledge can be attributed to three main factors, which summarize the complexity of the radioprotection field: 1. radiosensitivity is species-specific; 2. non-human biota cannot be reallocated in the event of radioactive contamination; and 3. the integrity of the affected ecosystem(s) depends on the interaction of all species within it. To improve our scientific knowledge and the background for decision-makers, it is essential to study the effects of IR exposure on different taxa. By expanding the comparison possibilities, it may be possible to identify common factors that would allow a better extrapolation of the expected outcomes to other organisms and higher ecological hierarchies. This can only be achieved through basic research for unravelling the mechanisms behind the responses and adaptation to IR, but also the synergistic effects with other stress factors.

Approaching it from a different point of view, IR is also an ancient ecological factor. Natural IR levels on Earth’s surface are relatively low for the last ca. 600 million years, and all contemporary organisms are well-adapted to it. As a primordial damaging stress factor, the mechanisms involved in IR defence are commonly found across life domains, from *Archaea* to *Eukarya*. Such an ancient evolutionary origin also reflects the fundamental importance of genome integrity preservation for life maintenance. Even though the role of IR in plant evolution remains elusive, it is expected to adjust evolutionary rates as a genotoxic factor. Indeed, considering the energy costs for the defence machinery, there must be a trade-off between protection responses and overall metabolic processes, ultimately dictating the genetic diversification and evolutionary pace. This hypothesis is supported by an interesting overlap between the reduction of surface IR levels during Earth’s evolution and the evolution of DNA repair gene families following the diversification of *Viridiplantae*. The decreasing IR exposure also represented a lower activity demand of the costly DNA repair system. This event probably created a gap for energetic optimization, which is represented by the depletion of DNA repair genes and by the emergence of more complex systems and species.

## Figures and Tables

**Figure 1 plants-12-01178-f001:**
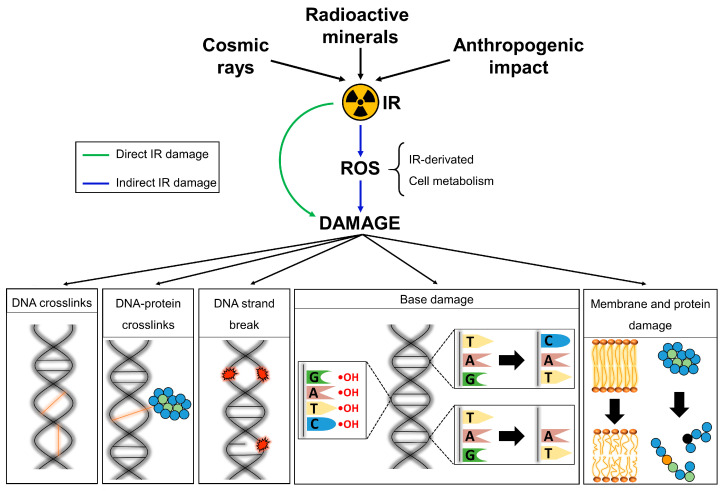
Overview of ionizing radiation (IR) sources and their effect on living organisms. IR can cause direct (green arrow) or indirect (blue arrows) damage to living organisms. The latter is a consequence of the generation of reactive oxygen species (ROS) due to the radiolysis of water by the ionizing high-energy particles or waves. Yet, ROS are also innate to cell metabolism and act as signalling molecules, meaning that upon IR exposure the cells undergo homeostatic imbalance.

**Figure 2 plants-12-01178-f002:**
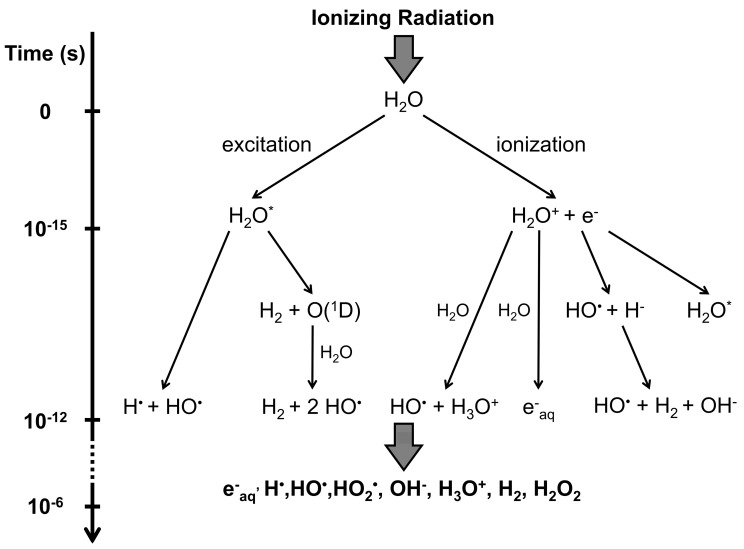
Scheme representing the IR-promoted water radiolysis steps, resulting in the generation of free radicals and ROS. H_2_O* represents excited water molecule. Adapted from [64].

**Figure 3 plants-12-01178-f003:**
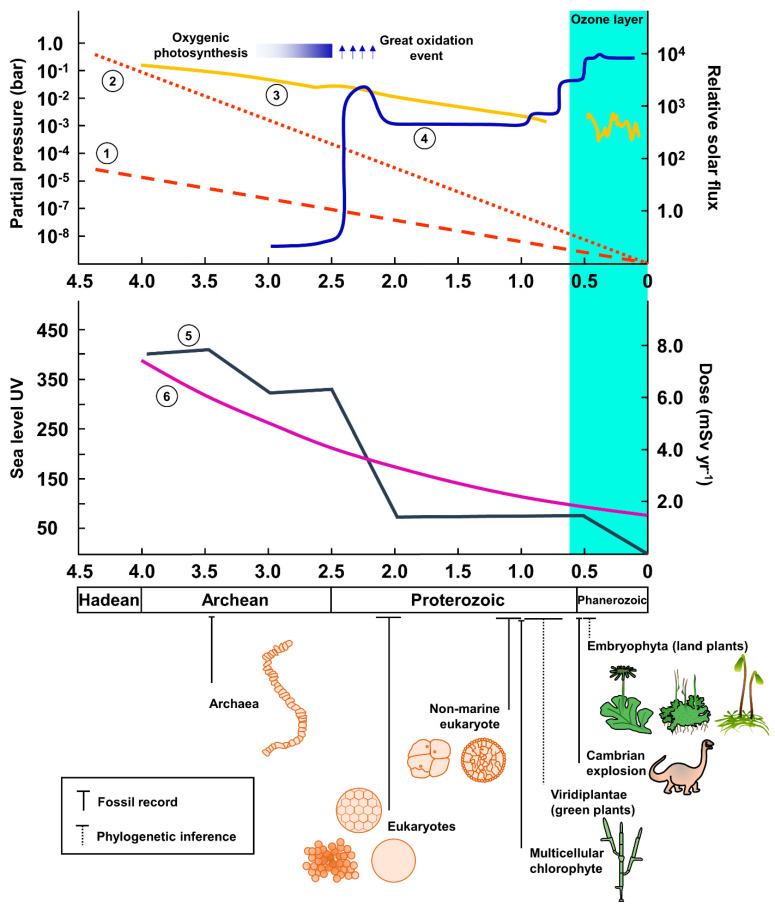
Environmental factors that may have contributed to the evolution of the plant DNA repair machinery. The gases’ partial pressure, relative solar flux, and sea level UV are in comparison to present time values. Both panels are plotted as a function of time in billions of years. Geological eras are represented at the bottom and include main life evolution events according to fossil records [196,231,232,233] and phylogenetic inference [207,208,209]. (**A**)**. *Curves 1*** (1–20 Å, high-energy) and ***2*** (920–1200 Å) represent solar normalized fluxes over time [15,189]. ***Curves 3*** and ***4*** represent the relative CO_2_ and O_2_ pressures over time [234,235]. The blue-shaded area represents the expected establishment of the shielding ozone layer ca. 600 Mya, which may have depended on the increase of Earth’s magnetosphere [15,226]. (**B**)**. *Curve 5*** represents the sea level UV flux over time, which had sharp drops attributed to atmosphere oxygenation [188]. ***Curve 6*** shows the changes in background radiation levels over time [188].

**Figure 4 plants-12-01178-f004:**
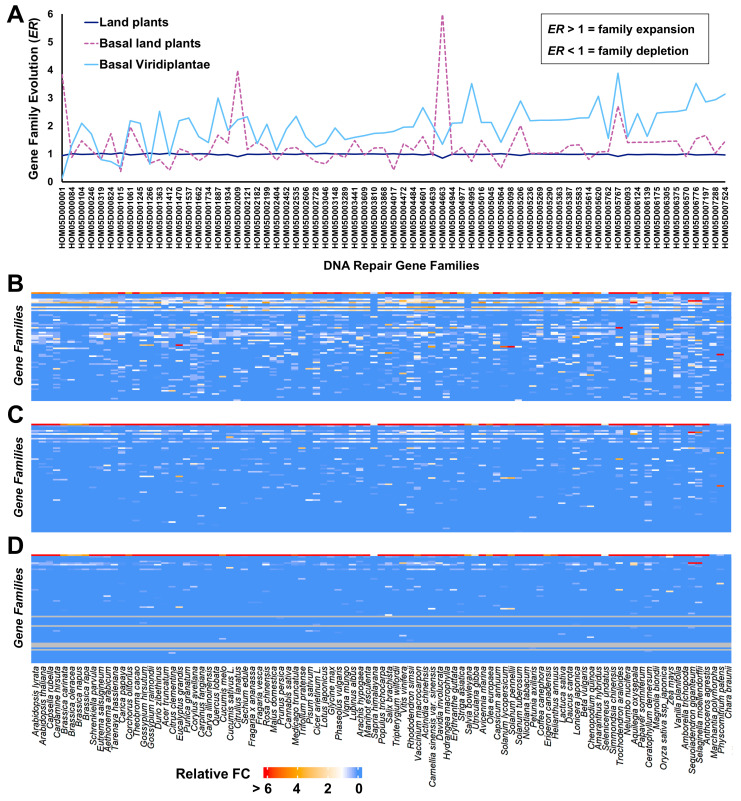
Evolution of DNA repair gene families in *Viridiplantae*. (**A**) Comparison of family expansion/depletion among basal *Viridiplantae* (*Chlorophyta* + *Prasinodermophyta*), basal land plants (*Lycopodiopsida* + *Anthocerotopsida* + *Marchantiopsida* + *Bryopsida*), and remaining land plants (*Magnoliopsida* + *Pinopsida*). The gene family evolution rate (*ER*) values are given in Appendix A, and calculation details are given in Appendix B. In summary, family expansion is represented by values > 1 and depletion by values < 1. The *ER* was assessed by the gene copy number of a given family compared to the proteome size of each species, normalized over all species. Gene families’ data were retrieved from PLAZA V5 (https://bioinformatics.psb.ugent.be/plaza/, accessed on 1 October 2021) and are provided in Appendix A. (**B**–**D**). For each DNA repair gene family (*y* axis), the relative fold change of the *ER* was calculated between each *Streptophyta* (vascular plants, *x* axis) and each basal *Viridiplantae*: *C. reinhardtii* (panel **B**), *M. commode* (panel **C**), and *P. colonial* (panel **D**). The representation of the gene families in the *y* axis (top-bottom) follows the same order as the *x* axis labels in panel (**A**) (left-right).

## Data Availability

All data used for DNA repair gene family evolution analysis is provided in Appendix A. Details about the data analysis is provided in Appendix B.

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
