# Peer review of "Chronic Ionizing Radiation of Plants: An Evolutionary Factor from Direct Damage to Non-Target Effects"

_plants, 2023, doi:10.3390/plants12051178_

Round 1

Reviewer 1 Report

Most of the studies on Ionizing radiations focused on their lethal impacts on living organisms. In the current review article, the authors discussed DNA damage by IR radiation, IR defense system and also discussed the evolutionary factor, plant diversification and territorialization. The article was well written. It would be nice if the authors includes a section future perspectives or if they give more importance to the last 6 lines of conclusion and discuss more.

Author Response

Dear reviewer,

We really appreciate your interest and feedback. Following your suggestion, we introduced a “Perspectives” section (lines 710-735), in which the ideas previously comprised in those lines in the conclusion have been expanded. All modifications made in the text are highlighted.

Sincerely,

Gustavo T. Duarte, on behalf of all authors.

Reviewer 2 Report

Comments

Thank you for sending me this manuscript to review. Overall, I judge the manuscript an interesting summary of different impacts that Ionizing radiation has on plants and its different pathways, including DNA maintenance, SOS response, DNA repair, etc.

Gustavo Turqueto Duarte, et al., describe molecular mechanisms of radiation responses in plants and the authors suggest that ionizing radiation played an important role in land colonization and plant diversification rates. The authors hypothesize that a decrease in levels of radiation exposure on the surface overlaps with a DNA repair  gene family’s depletion in land plants compared to ancestral groups.

There are minor and major issues that are listed in order, as follow:

1) The introduction is general, and some paragraphs do not connect the main idea in this review

2) in page 2, line 53-54: “ 1) in page 2, line 55: “It is plausible to suppose that the maintenance of cellular and genomic integrities was more critical in the past, thus requiring a higher activity of the energetically costly DNA repair systems” please explain or specify what part of the past the author are talking about, cause at the very beginning of organism evolution (bacteria mostly) the DNA repair systems were very primitive and some mechanisms had not appeared yet.

3) Section 2 on page 3. Add more information to the last paragraph in which is little explored the effect of radiation due to radionuclides on plant and hydrothermal vent ecosystem.

4) I think the impact of IR on plant mitochondrial and chloroplastic DNA repair and maintenance mechanisms  should be explored

5) the depletion of certain DNA repair gene families does not mean it was due to a drop of IR, it would be only a factor out of many others. Sometimes a depletion of certain gene families is due to a gain of function in other molecular mechanisms, response to a new environment conditions  (no only one).

Author Response

Dear reviewer,

We really appreciate your interest and feedback. Please find below the point-by-point response to your comments. All modifications made in the text are highlighted.

Sincerely,

Gustavo T. Duarte, on behalf of all authors.

1) The introduction is general, and some paragraphs do not connect the main idea in this review

After revising the text, we also found that the introduction required improvement. The idea of the introduction was to give a general overview of all the content discussed in the review, but indeed the reading flow was compromised. The introduction has been mostly modified for improving the connection of the ideas.

 2) in page 2, line 53-54: “ 1) in page 2, line 55: “It is plausible to suppose that the maintenance of cellular and genomic integrities was more critical in the past, thus requiring a higher activity of the energetically costly DNA repair systems” please explain or specify what part of the past the author are talking about, cause at the very beginning of organism evolution (bacteria mostly) the DNA repair systems were very primitive and some mechanisms had not appeared yet.

As part of the modifications in the introduction, this sentence has been improved. The idea is better developed in the last paragraph of the introduction. Considering that the levels of ionizing radiation were high since the origin of our planet, it is possible that the evolution of DNA defence mechanisms must have occurred rather early during evolution in order for the establishment of life (in the sense of a unit capable of maintaining and transferring its genetic information to the next generation). Indeed, some repair mechanisms are also found in Archaea. These defence systems would later diversify and gain complexity.

3) Section 2 on page 3. Add more information to the last paragraph in which is little explored the effect of radiation due to radionuclides on plant and hydrothermal vent ecosystem.

This is indeed an interesting point, which has been improved (lines 110-118). However, there is still few information available in the literature as it is a recent finding, and the depths at which the hydrothermal vents are found probably limit the investigations.

 4) I think the impact of IR on plant mitochondrial and chloroplastic DNA repair and maintenance mechanisms  should be explored

We agree with the reviewer that it is an interesting aspect of IR damage, although it was not the primary aim of this review. However, it seems that in general IR damage to organellar DNA still has been poorly explored, especially in the case of chloroplasts. We introduced some lines for presenting the organellar damage subject (section 4, lines 186-188), and also provided some information about their intrinsic defence mechanisms (section 5, lines 249-265).

 5) the depletion of certain DNA repair gene families does not mean it was due to a drop of IR, it would be only a factor out of many others. Sometimes a depletion of certain gene families is due to a gain of function in other molecular mechanisms, response to a new environment conditions  (no only one).

Indeed we cannot exclude the possibility that confounding factors and the gain of function of certain DNA repair gene families may have contributed to the depletion of other ones. This observation has been introduced in the text (lines 681-682). Nevertheless, the time correlation between the depletion of repair genes in Embryophyta with the decrease of IR levels on Earth is peculiar, especially when considering that the machinery that would allow land colonization was probably available millions of years before the event occurred (reference 212). In this context, IR would at least be the physical barrier limiting life on the surface. However, considering the energetic costs of the repair machinery (references 175 and 176), it is plausible to expect a decrease on the number of active elements over time, once the IR levels declined.

Reviewer 3 Report

  • The abstract is carelessly written. Authors should incorporate their notable findings and adequately connect them with the sentences they choose to correspond.   
  •  The introduction section must have a clear hypothesis and significantly develop the second paragraph of your manuscript. Make it more connected to the problem statement. 
  • Overall, there is the repetition of the information, which could be avoided.  
  • Use Italics with the scientific names.   
  • There is some repetition in the last paragraph of the discussion section. 
  • The conclusion section should be more precise and highlight only the findings of the present manuscript.
  •  It should be rewritten. Overall, it can be suitable for publication after incorporating suggestions.

Author Response

Dear reviewer,

We appreciate your suggestions. We rearranged the text in order to improve the reading flow, also following the recommendations given by the other reviewers. The introduction has been mostly modified for connecting the ideas discussed in the review, and part of the conclusion was used for developing a “Perspectives” section, following Reviewer’s 2 suggestion. The scientific nomenclature has been corrected. All modifications are highlighted.

Sincerely,

Gustavo T. Duarte, on behalf of all authors.